# Histo-Miner: Deep learning based tissue features extraction pipeline from H&E whole slide images of cutaneous squamous cell carcinoma

Lucas Sancéré[1,2,3]*, Carina Lorenz[3,4,5], Doris Helbig[6], Oana-Diana Persa[7],
Sonja Dengler[8], Alexander Kreuter[9], Martim Laimer[10], Roland Lang[10], Anne Fröhlich[11],
Jennifer Landsberg[11], Johannes Brägelmann[3,4,5,12]*, Katarzyna Bozek[2,3,13]*

**1** Faculty of Mathematics and Natural Sciences, University of Cologne, Cologne, North Rhine-Westphalia, Germany, **2** Institute for Biomedical Informatics, Faculty of Medicine and University Hospital Cologne, University of Cologne, Cologne, North Rhine-Westphalia, Germany, **3** Center for Molecular Medicine Cologne (CMMC), Faculty of Medicine and University Hospital Cologne, University of Cologne, Cologne, North Rhine-Westphalia, Germany, **4** University of Cologne, Faculty of Medicine and University Hospital Cologne, Department of Translational Genomics, Cologne, Germany, **5** University of Cologne, Faculty of Medicine and University Hospital Cologne, Mildred Scheel School of Oncology, Cologne, Germany, **6** Department for Dermatology, University Hospital Cologne, Cologne, Germany, **7** Department of Dermatology and Allergy, School of Medicine, Technical University of Munich, Bavarian Cancer Research Center (BZKF), Munich, Germany, **8** Department of Dermatology, Dortmund Hospital gGmbH, University Witten/Herdecke, Dortmund, Germany, **9** Department of Dermatology, Venereology and Allergology, Helios St. Elisabeth Hospital Oberhausen, University Witten/Herdecke, Oberhausen, Germany, **10** Department of Dermatology and Allergology, University Hospital of the Paracelsus Medical University Salzburg, Salzburg, Austria, **11** Department of Dermatology and Allergology, University Hospital Bonn, Bonn, Germany, **12** Medical Clinic III for Oncology, Hematology, Immune-Oncology and Rheumatology, University Hospital Bonn (UKB), Bonn, Germany, **13** Excellence Cluster on Cellular Stress Responses in Aging-Associated Diseases (CECAD), University of Cologne, Cologne, North Rhine-Westphalia, Germany

☙ These authors contributed equally to this work.
* lsancere@uni-koeln.de (LS); johannes.braegelmann@uni-koeln.de (JB); k.bozek@uni-koeln.de (KB)

## Abstract

Recent advances in digital pathology have enabled comprehensive analyses of Whole-Slide Images (WSIs) from tissue samples, leveraging high-resolution microscopy and computational capabilities. Despite this progress, available tools for automatic cell type identification perform poorly on skin tissue, e.g. in the classification of non-melanoma tumor cells. This is due to a paucity of labeled training data sets and high morphological similarities between tumor and non-tumor epithelial cells in the skin. Here, we propose Histo-Miner, a deep learning-based pipeline designed for the analysis of skin WSIs. To this end we generated two new datasets using WSIs of cutaneous Squamous Cell Carcinoma (cSCC) samples, a frequent non-melanoma skin cancer, by annotating 47,392 cell nuclei across 5 cell types in 21 WSIs and segmenting tumor regions in 144 WSIs. Histo-Miner employs convolutional neural networks and vision transformers for nucleus segmentation and classification, as well as tumor region segmentation. Performance of trained models positively compares to state of the art with multi-class Panoptic Quality (mPQ) of 0.569 for nucleus segmentation, macro-averaged F1 of 0.832 for nucleus classification and mean

**Data availability statement:** *Data availability*: The checkpoints of SCC Hovernet and SCC Segmenter are available on Zenodo repository https://doi.org/10.5281/zenodo.13970197. TumSeg and NucSeg datasets are also available on Zenodo repository https://doi.org/10.5281/zenodo.8362592. The CPI image classification dataset utilized as a use-case for Histo-Miner is also available on Zenodo repository https://doi.org/10.5281/zenodo.13986859. Finally, TumSeg test set, SCC Segmenter inference on test set, the list of all features from Tissue Analyser and ranking of features after cross-validation with mRMR selection are available on Zenodo repository https://doi.org/10.5281/zenodo.15836084. All the WSIs of these datasets were anonymized and cannot be used for commercial use. *Code availability*: Our tool is available through our github repository: https://github.com/bozeklab/histo-miner.

**Funding:** This work was supported by the Ministry for Culture and Science (MKW) of the State of North Rhine-Westphalia [grant number 311-8.03.03.02-147635]. LS and KB were supported by the North Rhine-Westphalia return program (311-8.03.03.02-147635) and hosted by the Center for Molecular Medicine Cologne. AF was partly funded by the Deutsche Krebshilfe through a Mildred Scheel Foundation Grant (grant number 70113307). CL was partly funded through the collaborative research center grant on small cell lung cancer (CRC1399, project ID 413326622) and a project grant (grant ID BR 6949) by the German Research Foundation (DFG). JB receives funding through the collaborative research center grant on small cell lung cancer (CRC1399, project ID 413326622) and on predictability in evolution (CRC1310, project ID 325931972) by the German Research Foundation (Deutsche Forschungsgemeinschaft, DFG), a Mildred Scheel Nachwuchszentrum Grant 70113307 and project funding (grant ID 70116929) by the German Cancer Aid

Intersection over Union (mIoU) of 0.907 for tumor region segmentation. From these output, the pipeline can generate a compact feature vector summarizing tissue morphology and cellular interactions, which can be used for various downstream tasks. As an exemplary use-case, we deploy Histo-Miner to predict cSCC patient response to immunotherapy based on pre-treatment WSIs from 45 patients. Histo-Miner predicts patient response with mean area under ROC curve of $0.755 \pm 0.091$ over cross-validation, and identifies percentages of lymphocytes, the granulocyte to lymphocyte ratio in tumor vicinity and the distances between granulocytes and plasma cells in tumors as predictive features for therapy response. This highlights the applicability of Histo-Miner to clinically relevant scenarios, providing direct interpretation of the classification and insights into the underlying biology. Importantly, Histo-Miner is designed to allow for its use on other cancer types and on other training datasets.

## Author summary

Digital pathology is transforming how we study disease by turning tissue samples into high-resolution images that capture the architecture of entire tumors. However, these images are vast and complex, making it difficult to extract meaningful clinical insights without advanced computational tools. In this work, we present Histo-Miner, a framework designed to systematically analyze these images at multiple levels of detail—from one single cell to entire tissue regions. We apply this approach to cutaneous squamous cell carcinoma, a common form of skin cancer, demonstrating how large-scale tissue data can be mined for biological insights. Our method identifies and characterizes different types of cells, maps how they are organized within tumor areas, and connects these patterns to patient outcomes. Through this lens, we uncover subtle features of the tissue environment that may influence how patients respond to therapy. We find that the most informative features describe the presence and balance of different types of immune system cells, and how these cells are spatially arranged within the tissue. Beyond its immediate findings, Histo-Miner, provides openly available data and tools that aim to make large-scale tissue analysis more interpretable, reproducible, and transferable to other diseases.

## Introduction

Digital pathology slide scanners and advancements in computer vision allow for automation of diagnostic pathology tasks. Hematoxylin and Eosin (H&E) staining [1] is widely used in pathology and represents a standard that both the classical and digital pathology are based on. The resulting tissue scans are called Whole-Slide Images (WSIs). Given the large size of WSIs containing thousands to millions of cells, automated methods for WSI analysis are indispensable to systematically and comprehensively quantify their content. A large panel of tasks can be performed by machine learning and deep learning models on this type of images: segmentation of nuclei and tumors in the WSIs [2,3], image classification [4], or discovery of new

(Deutsche Krebshilfe) and the CANTAR network (NW21-062B) funded through the program "Netzwerke 2021", an initiative of the Ministry of Culture and Science of the State of Northrhine Westphalia, Germany. The funders had no role in study design, data collection and analysis, decision to publish, or preparation of the manuscript.

**Competing interests:** I have read the journal's policy and the authors of this manuscript have the following competing interests: JB has received research funding from Bayer AG and travel expenses from Merck KG & Bicycle Therapeutics and serves as a consultant for Bicycle Therapeutics outside of the presented work. These entities had no role in study design, data collection and analysis, publication decisions, or manuscript preparation. ODP received personal honoraria from Merck Sharp & Dohme and Almirall, received travel support from Kyowa Kirin, Pierre Fabre, Sanofi and Sun Pharma outside of the presented work and is member of the advisory board of Bristol Myers Squibb and Sanofi. These entities had no role in study design, data collection and analysis, publication decisions, or manuscript preparation. All other authors declare no conflicts of interests.

biomarkers [5]. Importantly, such methods automate time consuming intermediary tasks, such as cell counting, that allows the practitioner to focus on diagnosis and interpretation [6,7].

While there is a range of datasets and methods in digital pathology [8–11], few of them are dedicated to skin and non-melanoma skin tumors. Skin differs from other tissues in its unique structure, composition, and function, which presents specific challenges for digital pathology methods. The skin consists of multiple distinct layers, including the epidermis, dermis, and subcutaneous tissue, each with varying cell types, densities, and extracellular matrices. These variations lead to textural patterns and coloration that are unique to WSIs of skin. Therefore, specialized methods tailored to the unique characteristics of skin tissue are necessary for reliable digital pathology in dermatology.

Here, we focus on cutaneous squamous cell carcinoma (cSCC) - the second most common form of non-melanoma skin cancer in the USA and widely spread worldwide [12]. While the majority of cSCCs can be cured by surgery alone, 5-10% of cSCC patients experience disease recurrence or metastases, requiring systemic treatments [13]. While the effects of chemotherapy are very limited, systemic treatment with immune checkpoint inhibition (CPI) has emerged as a promising alternative. However, up to half of patients do not respond to the immunotherapy. To date, it is impossible to predict, which patients have a high chance of response to CPI and which patients may require other/additional treatment modalities [13]. Quantitative methods for analysis of cSCC patient samples would provide more insights into the morphological variability of this tumor type and could potentially allow for identification of morphological markers linked to the patient risk of progression.

We propose a deep learning-based pipeline, Histo-Miner with openly available code and datasets, for development and analysis of cSCC samples. We generated a dataset called NucSeg containing manually annotated class labels and segmentation masks of 47,392 cell nuclei from 21 WSIs of cSCC. Furthermore, we generated TumSeg dataset, containing binary segmentation masks of tumor regions in 144 WSIs of cSCC. Our pipeline performs segmentation and classification of cell types into 6 different classes (granulocytes, lymphocytes, plasma cells, stromal cells, tumor cells and epithelial cells) and tumor region segmentation, both using deep learning models trained on our datasets. Histo-Miner uses the segmentation and classification results to encode WSIs of cSCC into features describing tissue morphology, organization, and cellular interactions. The code is open-source, customizable, and each part (tumor segmentation, cell type identification) can be used separately to fit user needs. The training datasets, as well as the models weights used in the intermediary steps are publicly available (See **Data availability statement**).

We finally tested our Histo-Miner pipeline to predict cSCC patient response to immunotherapy. Immunotherapy with anti-PD1 antibodies is the major treatment for patients with advanced cSCC, but currently no predictive biomarkers are established to identify patients with a high likelihood of therapy failure. We generated the CPI dataset including 45 skin WSIs of 45 patients, before they received immunotherapy treatment, and annotated these slides as responder or non responder to the treatment. Using the features produced by Histo-Miner we classified patient response

and found interpretable features explaining model choices. These features provide insights into biological factors favoring treatment response. The CPI dataset and the feature list are publicly available (See **Data availability statement**).

## Materials and methods

### Ethics statement

The study was performed in agreement with the Declaration of Helsinki Institutional Review Board of the University Hospital Bonn (vote number 187/16), Ethics committee of the University Hospital Cologne (votenumbers 21–1500, 20–1082 and 22–1330-retro) and institutional review board of the TU Munich (vote number 2024–363-S-CB - 1). Need for informed consent was waived for this retrospective analysis using anonymized data.

### Histo-Miner pipeline description

To describe the tissue organization and composition of cSCC, and obtain detailed information on the histomorphology of these tumors we developed our pipeline, Histo-Miner (Fig 1). It uses both cell nuclei and tumor segmentation as first steps to quantitatively describe tumor sample morphology.

In the pre-processing pipeline of SCC Segmenter, the images are downsampled, tiled into patches and the patches are normalized using mean and standard deviation of RGB pixel values of ImageNet 1K (see our github repository for implementation details). SCC Hovernet is trained with data augmentation for model generalization [14] and then input patches don't need color normalization. To capture cellular heterogeneity, the cell nuclei are segmented and classified into: granulocytes, lymphocytes, plasma cells, stromal cells, and tumor cells. Cell nuclei segmentation and classification is performed with Hovernet convolutional network [14] trained on a manually annotated set of cSCC WSIs (NucSeg). This model shows better instance segmentation and improved Panoptic Quality (PQ) compared to other recent H&E segmentation models [14,15]. The model is open source and allows users to train and adapt it. Our pipeline additionally includes Segmenter vision transformer network to segment tumor regions. This vision transformer model outperforms other models in several benchmark tasks of instance and semantic segmentation [16–18]. We trained the Segmenter model on our TumSeg dataset to perform a binary pixel-wise classification tumor and non-tumor regions of the same WSIs that were used for Hovernet inference. We name the resulting trained networks SCC Hovernet and SCC Segmenter.

In the pre-processing pipeline of SCC Segmenter, the images are first downsampled, then tiled into patches and the patches are normalized using mean and standard deviation of RGB pixel values of ImageNet 1K (see our github repository for implementation details). In the pre-processing pipeline of SCC Hovernet, WSI input are only tiled into patches. In both cases, only patches with tissue are kept for prediction (at least one pixel of tissue) after tiling.

After determining tumor areas using SCC Segmenter, the results of the SCC Hovernet cell nuclei classification are updated to add a new cell class as follows: all the nuclei predicted as tumor cells outside of the predicted tumor regions are reclassified as healthy epithelial. The reason for this update is that healthy epithelial cells and tumor cells have similar morphologies and are impossible to discriminate without a broader context and information about the tissue structure (see Fig A in S1 Material). Example visualization of the inference of the two methods is shown in Fig 2.

Results of segmentation are input to tissue analysis part of our pipeline. In this part we perform calculation of 317 features that describe and encode the tissue samples. Example features include percentages of cells of specific class anywhere in the sample as well as inside the tumor regions. For every pair of cell classes X and Y, we also calculate the average distance of the closest cell of class Y to a cell of class X inside the tumor regions. This feature describes the topology of the tissue and the interactions between cell classes.

An exhaustive list of the calculated features is available in S1 Material. We do not consider absolute numbers of cells of a given class as a feature, but cell densities and percentages, as this metric is dependent on the WSI size and not the structure of the tissue itself. All the features are stored in a light json file, which results in encoding and compressing a WSI of multiple GB into a text file of 3.7 KB. These features are a convenient WSI representation for any downstream

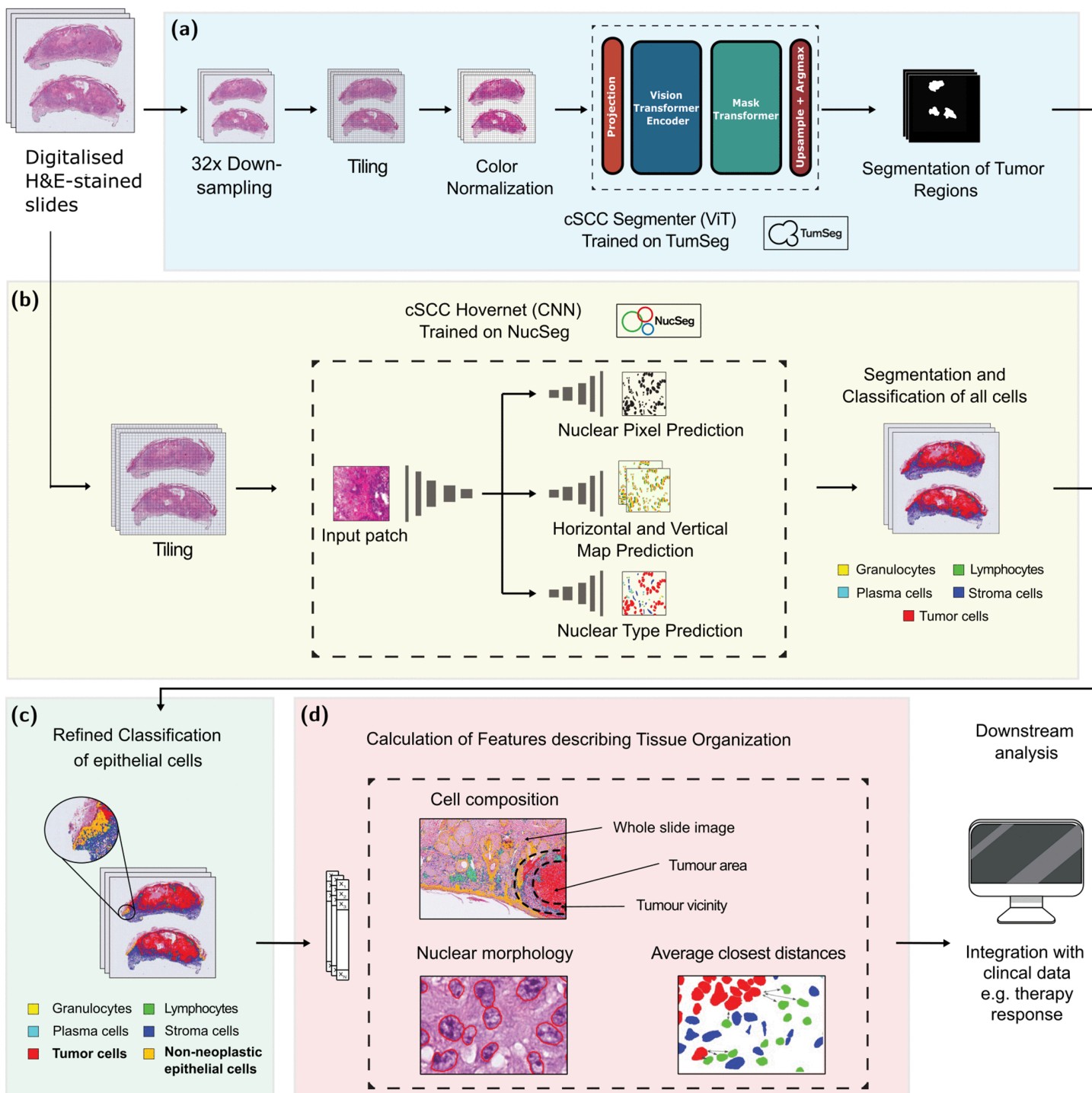

**Fig 1**. **Overview of Histo-Miner pipeline.** The pipeline uses WSI from cSCC patient as input. **(a) & (b)** During inference, WSI images are tiled into patches and undergo pre-processing pipelines (see **Methods**). After pre-processing, SCC Segmenter performs tumor region binary segmentation on processed patches and SCC Hovernet segments and classifies cell nuclei. **(c)** Using the output of SCC Segmenter, cell classification is refined by adding a new cell class: non-neoplastic epithelial. This last result is saved in a json text file. A visualization of resulting annotations is provided in Fig 2. **(d)** Using refined segmentation and classification of nuclei, together with segmentation of tumor regions, we calculate features that describe the tissue organization. Example features include e.g. percentage of lymphocytes in the vicinity of the tumor and average closest distance between tumor and lymphocyte cells. A list of all 317 calculated features is provided in S1 Material.

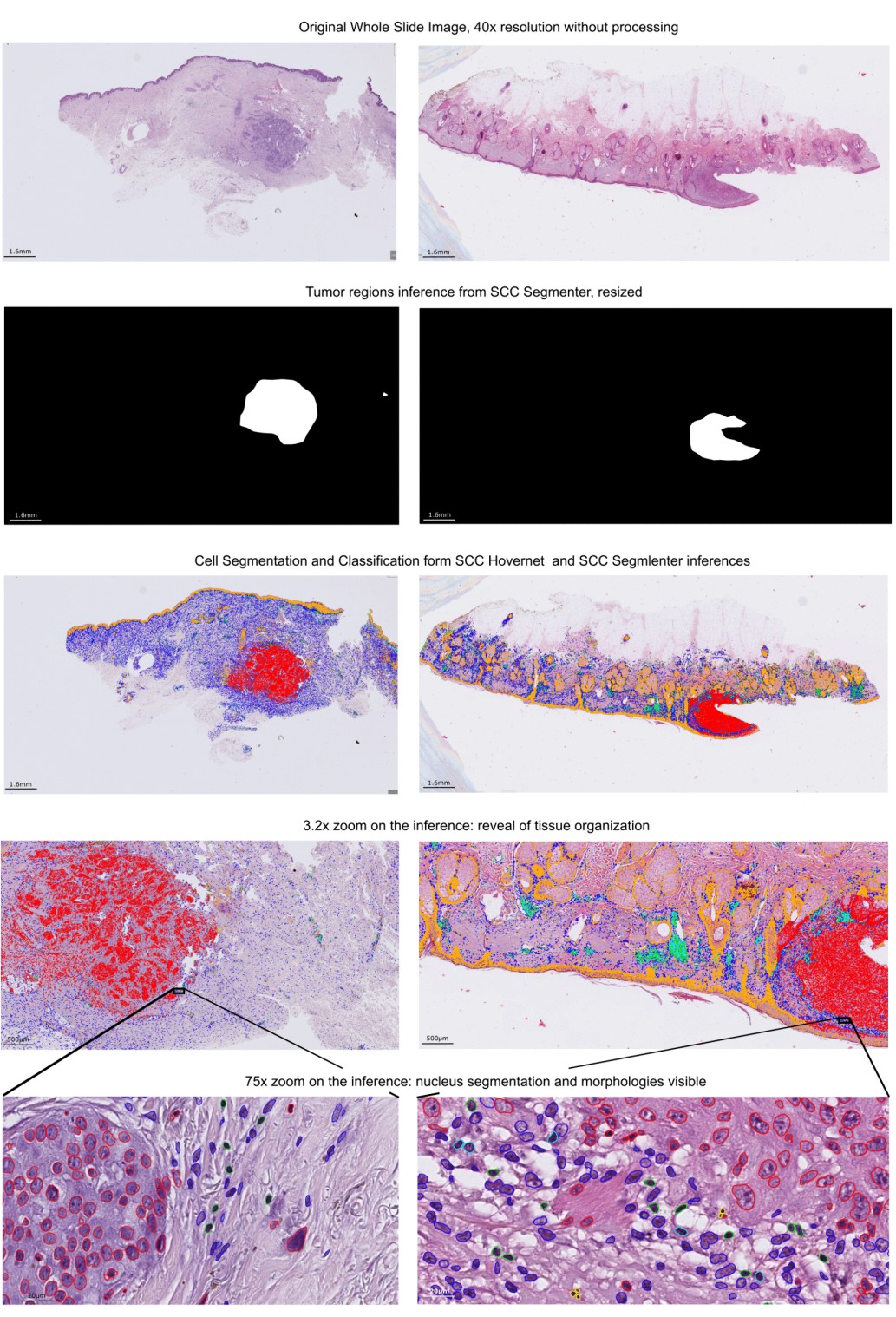

**Fig 2. Predictions of SCC Hovernet and SCC Segmenter models.** The different images correspond to different steps of the Histo-Miner pipeline as depicted in Fig 1. 2 WSIs of 2 different patients from 2 different cohorts are shown. The H&E staining differs between the slides showing varying hues of blue and pink. After predicting the tumor area with SCC Segmenter, Histo-Miner segments and then classifies cells into five different classes:

granulocytes, lymphocytes, plasma cells, stromal cells, tumor cells. Using tumor segmentation, tumor cells detected outside tumor regions are re-classified as non-neoplastic. The cell nuclei segmentation and classification illustrate sample organization at tissue level (3x zoom), or at cell level (75x zoom). Based on segmentation results Histo-Miner calculates features describing cell-level and tissue-level tumor organization. Also in the case of damaged sample (one part of the tumor is missing in the WSI on the right), the model is not hallucinating segmentation of the remaining parts of the sample.

analysis. All the different steps of the pipeline can be run separately as well as configured to fit specific needs. An example of use case, predicting response of cSCC patients to immunotherapy, is provided in the following sections.

### NucSeg and TumSeg datasets descriptions

To enable segmentation and cell nucleus type classification for cSCC, we assembled 21 WSIs of H&E-stained tissue sections of 20 cSCC patients from the University Hospital Cologne. The images were acquired using a NanoZoomer Slide Scanner (Hamamatsu). In the images the nuclei contours were marked and assigned to five cell types: granulocytes, lymphocytes, plasma cells, stromal cells, and tumor cells. 1,707 H&E non-overlapping patches of 256x256 pixels, with 40x and 20x resolutions, have been manually annotated by two pathology experts. To ensure annotation consistency across the distributed workflow, ambiguous morphological patterns were subjected to joint consensus review and compared to IHC staining on validation slides. 47,392 nuclei were labeled (classified and segmented) in total (3,135 granulocytes, 12,263 lymphocytes, 3,271 plasma cells, 11,526 stromal cells, 17,197 tumor cells), see Fig 3a and Fig 3b.

### Histo-Miner deep learning models

Histo-Miner implementation includes deep learning models trained with our custom datasets. Histo-Miner users can utilize the weights of these models to perform similar inferences on their own datasets, re-train these models through Histo-Miner implementation directly and edit model architectures and hyperparameters for further development. To fit user needs, it is possible to use only specific blocks of the pipeline - such as inference of the deep learning models - instead of using the whole process until calculation of tissue features.

We performed segmentation and classification of segmented cell nuclei using Hovernet model [14], which we selected based on its performance and ease of use. The model is a convolutional neural network containing encoder and decoder parts. The semantic segmentation of tumor region on the WSIs was achieved using Segmenter, a vision transformer model [16]. Segmenter is a collection of architectures with varying size, composed of encoder and decoder parts. In Histo-Miner pipeline we use the Seg-L-Mask/16 segmenter variant, achieving better results than the base model but requiring more GPU memory.

### Training SCC Hovernet

To segment and classify cell nuclei into different cell classes (granulocytes, lymphocytes, plasma cells, stromal cells, and tumor cells) we trained Hovernet network with our dataset NucSeg. The training was performed on 2 80GB A100 NVIDIA GPUs (Ampere micro-architecture). We used Hovernet with the encoder pretrained on ImageNet 21k. A second pre-training of 150 epochs was performed on a not-curated dataset of H&E nucleus segmented and classified, also made openly available. This dataset resemble NucSeg, consists of the same classes, but the segmentation and classification were mostly automatized and not fully corrected by human experts. During the first 50 epochs, the encoder weights were frozen and during the following 100 epochs all weights were updated. Finally, the main training of 250 epochs (first 50 epochs with frozen encoder weights followed by 200 epochs with all weights updated) was performed on NucSeg dataset. We used Adam optimization algorithm [19] with initial learning rate $\gamma_0$ of $10^{-4}$ which was reduced to $10^{-5}$ after 25 epochs. We used the same loss functions as in the original Hovernet model. The global loss function is an addition of

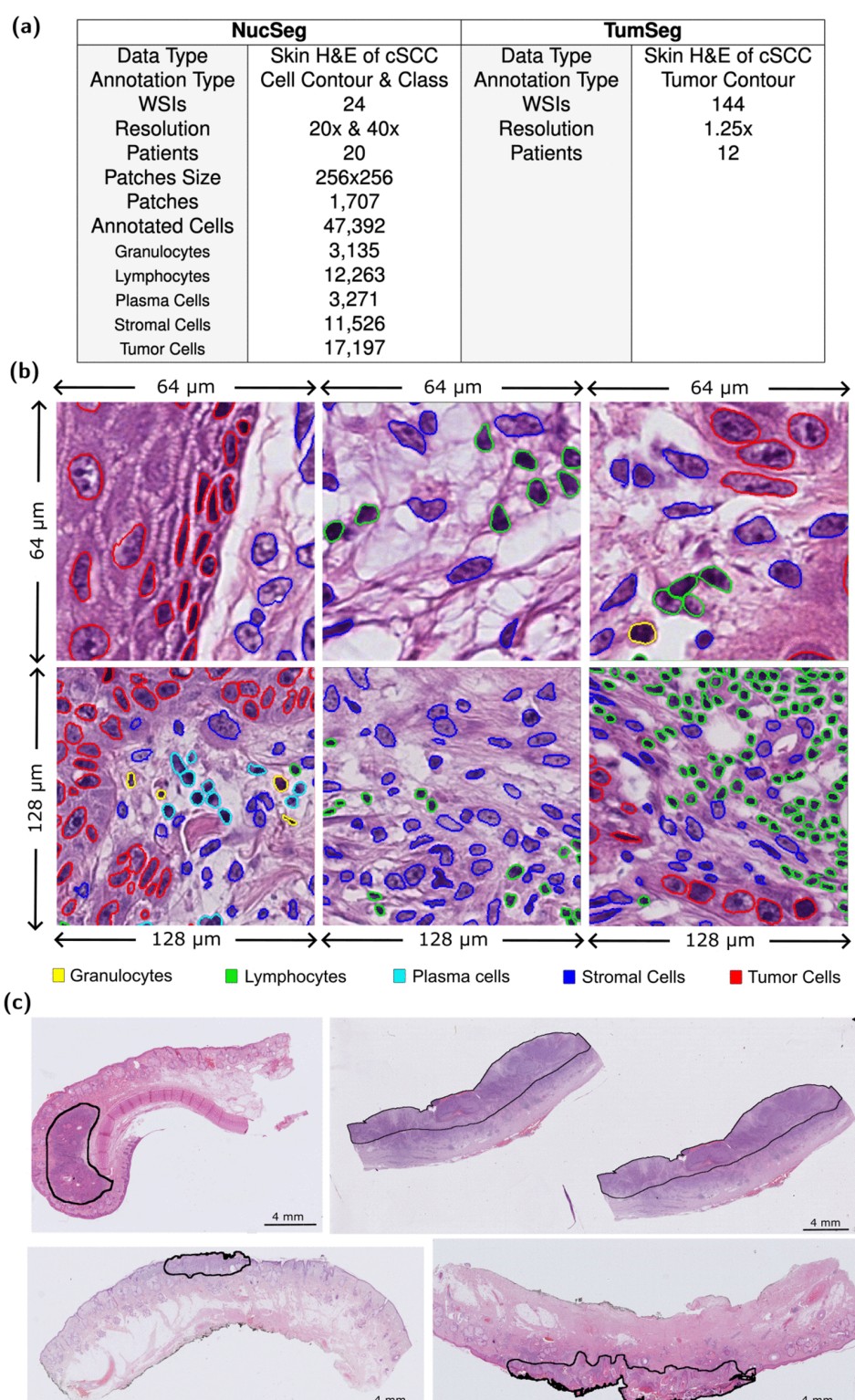

**(a)**

| | NucSeg | | TumSeg | |
|---|---|---|---|---|
| Data Type | Skin H&E of cSCC | Data Type | Skin H&E of cSCC | |
| Annotation Type | Cell Contour & Class | Annotation Type | Tumor Contour | |
| WSIs | 24 | WSIs | 144 | |
| Resolution | 20x & 40x | Resolution | 1.25x | |
| Patients | 20 | Patients | 12 | |
| Patches Size | 256x256 | | | |
| Patches | 1,707 | | | |
| Annotated Cells | 47,392 | | | |
| Granulocytes | 3,135 | | | |
| Lymphocytes | 12,263 | | | |
| Plasma Cells | 3,271 | | | |
| Stromal Cells | 11,526 | | | |
| Tumor Cells | 17,197 | | | |

**(b)**

Granulocytes   Lymphocytes   Plasma cells   Stromal Cells   Tumor Cells

**(c)**

**Fig 3**. **Visualization of samples from NucSeg and TumSeg training datasets. (a)** Overview of both datasets **(b)** Visualization of NucSeg training dataset. 47,392 cell nuclei from 1,707 H&E non-overlapping patches were segmented and classified. **(c)** Visualization of TumSeg training dataset. Tumor region are segmented by 2 experts on 144 WSIs.

three losses for each of the three branches of the Hovernet Model. These branches account, respectively, for the nuclear pixel classification task, the binary segmentation of nucleus, and the horizontal and vertical distance map used to separate touching instances [14]. We describe the loss function in detail in Eqs 1–4 in S1 Material. We used the following data augmentations during training: image flips, rotations, Gaussian blurs and median blurs according to the original Hovernet implementation.

We trained NucSeg dataset including 5,968 patches of size 540x540 pixels with 70% overlap. Our validation set contained 848 patches of size 540x540 pixels with 70% overlap. We kept the same set of hyperparameters as the original implementation [14], except that we doubled the training batch size for our last training step of 200 epochs (see above). We tested different training strategies combining network pre-trained on ImageNet 21k, pre-trained on the not-curated H&E nucleus dataset, and trained from scratch. Performance of resulting models was highest when the model was first pre-trained on ImageNet 21k, then pre-trained on the not-curated H&E nucleus dataset before fine-tuning. The choice of final model was based on maximizing the panoptic segmentation task performance.

### Training SCC segmenter

We trained Segmenter model with our dataset TumSeg. The training was performed on 2 80GB A100 NVIDIA GPUs (same as for Hovernet training). We used Vision Transformer pre-trained on ImageNet 21k [20] and fine-tuned it on our dataset TumSeg. We followed the data augmentation pipeline from the semantic segmentation library MMSegmentation [21]. It consists of random resizing of the image to a ratio between 0.5 and 2.0, random left-right flipping, and normalization of the images based on mean and standard deviation of pixel values of ImageNet 1k. We tested other normalization strategies, e.g taking mean and standard deviation of training dataset which resulted in worse model performance. We trained for 1,786 epochs (50,000 iterations) using Stochastic Gradient Descent as optimization method with learning rate following a polynomial decay scheme. Considering $\gamma$ the learning rate at the current iteration number, and $\gamma_0$ the base learning rate, the decay is defined as $\gamma = \gamma_0(\frac{1-N_{iter}}{N_{total}})^{0.9}$ where $N_{iter}$ and $N_{total}$ represent the current iteration number and the total iteration number, respectively. We set $\gamma_0$ to $10^{-3}$. We used cross-entropy without weight re-balancing as the loss function.

The model was trained on randomly chosen 115 slides of TumSeg dataset and the validation set contained the remaining 29 slides of the dataset. We performed hyperparameter grid-search to find the best set of hyperparameters as described in Table A in S1 Material. The accuracy estimation was performed on the validation set due to the limited number of slides and lack of an independent test set.

### Tissue analyser

Within Histo-Miner, SCC Segmenter model segments tumor regions. SCC Hovernet model performs instance segmentation of cell nuclei and classifies them into different cell classes (granulocytes, lymphocytes, plasma cells, stromal cells, and tumor cells). Using both models' predictions, we refine the cell classification and add one more class among the possible predictions, the healthy epithelial class. In fact, healthy epithelial cells and tumor cells are hard to discriminate without a broader context and information about the tissue structure. Healthy epithelial cells and tumor cells have similar morphologies. To distinguish these two cell types, we added one refinement step: all the nuclei predicted as belonging to tumor cells by SCC Hovernet, located outside the tumor regions predicted by Segmenter, are reclassified as epithelial. The result of the classification update is visible in Fig 2.

The updated cell nuclei segmentation and classification as well as the tissue segmentation are input to our Tissue Analyser part of the pipeline. Here we calculate 317 features capturing various aspects of tissue morphology and spatial organization. The features (see S1 Material) include: percentages of given cell types, composition of the tumor margin, ratio between cell types, repartition of a given cell type outside, inside and within the tumor margin. The final feature vector is a light but information-rich representation of the cSCC WSI for further downstream analyses.

One of our features is the average distance of the closest cell of a given class X - source class - to the closest cell of a given class Y - target class - inside the tumor regions. The average distance is calculated as shown in Eq 1 and through the following steps: **1-** We generate a rectangle that defines the initial search area. The rectangle height is 5% of the tumor bounding box height and width 5% of the tumor bounding box width. It is centered around the nucleus of the source cell class. **2-** We verify if there is at least one nucleus of the target cell class inside this search area. **3-** If there is at least one, we calculate all distances between source and target cells and keep the smallest one. If there is no nucleus of target cell class we increase the search area until we find at least one nucleus of target cell class in the search area. The search area cannot extend to other tumor areas. **4-** We perform steps 1-3 for all nuclei of the source class. A quantitative explanation is available in Algorithm 1 (all the memory optimization steps are skipped for readability). The increase of search area for each iteration was optimized to reduce calculation time. For computation optimization reasons, the search area is a rectangle and not a circle. Indeed, one of the main reason is that searching for coordinates in a rectangle is faster than searching for coordinates in a circle (only comparisons instead of subtractions and multiplications). In some specific cases, using a rectangle search area can lead to overestimation of the distance. Description of these cases, probability of overestimation, and bounding of overestimation are described in Fig C in S1 Material. This probability of overestimation decreases drastically with the number of cells in the tumor. For instance, we can calculate that for a squared search area of side length $2r_1$ and origin 0, if $N$ cells are in the circle of radius $\sqrt{2}r_1$ and origin 0 (the square is inscribed in this circle), the probability of overestimation is $P(error_N)_2 = 0.022$ for $N = 2$ and $P(error_N)_{10} = 2.8 \times 10^{-4}$ for $N = 10$. These distances describe the interactions between different cell types inside the tissue. They are calculated for granulocytes, lymphocytes, plasma cells, and tumor cells to assess the organization of the tissue regarding the intensity of the immune response.

$$\bar{d}_{closest_{c_A, c_B}} = \frac{1}{n_{c_A}} \sum_{(x_i, y_i) \in E_{c_A}} \min \left( \sqrt{(x_i - x_k)^2 + (y_i - y_k)^2} \right), \tag{1}$$

$$\forall (x_k, y_k) \in \left( E_{c_B} \cap E^{\lambda_0}_{(x,y) \in \mathcal{N}^2 |} \begin{cases} 0.05\,\lambda_0 l_t - x_i \le x \le 0.05\,\lambda_0 l_t + x_i \\ 0.05\,\lambda_0 w_t - y_i \le y \le 0.05\,\lambda_0 w_t + y_i \end{cases} \right)$$

with $\lambda_0$ verifying:

$$\forall \lambda \in \mathcal{N}^* \,|\, \left( E_{c_B} \cap E^{\lambda}_{(x,y) \in \mathcal{N}^2 |} \begin{cases} 0.05\,\lambda l_t - x_i \le x \le 0.05\,\lambda l_t + x_i \\ 0.05\,\lambda w_t - y_i \le y \le 0.05\,\lambda w_t + y_i \end{cases} \right) \ne \{\emptyset\}, \; \lambda_0 = \min(\lambda)$$

where $\lambda \in \mathcal{N}^*$ coefficient of increase of sides length of search rectangle, $l_t$ and $w_t$ length and width of the bounding box of the tumor region, $E_{c_a}$ set of coordinates in $\mathcal{N}^2$ of all cells centroid of class A (source class), $E_{c_b}$ set of coordinates in $\mathcal{N}^2$ of all cells centroid of class B (target class), $n_{c_A}$ number of cells of class A. $E^{\lambda}$ is the set of points inside the search rectangle of coefficient $\lambda$ and $E^{\lambda_0}$ is the set of points inside the smallest search rectangle that contains at least one centroid of cell of class B.

Other features include ratios of cell types and percentages of cells of a given cell type in the vicinity of the tumor, in the tumor regions or in the whole WSI. The vicinity of tumor is defined as a 1mm-wide area around the tumor [22]. In the case of ratios, to limit outlayers we define ratio $\eta = \frac{\log(n_{c_A}) + \epsilon}{\log(n_{c_B}) + \epsilon}$. with $n_{c_A}$ number of cells of class A, $n_{c_B}$ number of cells of class B and $\epsilon = 10^{-3}$, smoothness parameter. The full list of features is in the S1 Material.

**Algorithm 1** Minimum Distance Calculation Pseudo-Code

**Require:** *classjson*: file with centroid and type of all WSI's cells

**Require:** *maskmap*: segmentation of WSI's tumor regions

**Require:** *selected_classes*: classes for which we want to calculate distances, *f*: acceleration of increase parameter - here $f = 0.05$

Start $C(n, 2) = \frac{n(n-1)}{2}$ parallel processes, $n = \textbf{len}(selected\_classes)$

In each process a different pair source class / target class is defined

**For each** process:

 *allmindist* = **list()**

 **For all** cells of source class:

 *dist_list* = **list()**

 Check tumor ID of source cell

 List all target class cells *centroid coordinates* in the same Tumor

 Calculate length $l_t$ and width $w_t$ of the bounding box of the Tumor

 $\lambda = 1$

 *kept_targets* = **list()**

 **While** *kept_targets* cells list is **empty:**

 Construct search zone around source cell with $length = f\lambda l_t$ and $width = f\lambda w_t$

 Append to *kept_targets* **all** target cells *centroid coordinates* inside the search area

 $\lambda = \lambda + 1$

 **For all** cell in *kept_targets*:

 $dist = \sqrt{(x_i - x_k)^2 + (y_i - y_k)^2}$ with $(x_i, y_i)$ coordinates of cells of source class and $(x_k, y_k)$ coordinates of cells of target class

 *dist_list*.**append**(*dist*)

 *min_dist* = **min**(*dist_list*)

 *allmindist*.**append**(*min_dist*)

 Put item *avgdist* = **sum**(*allmindist*)/**len**(*allmindist*) in process queue

Ouput a list of all *avgdist* gathered items

## Feature selection to predict response of cSCC patients to immunotherapy

We collected WSIs from 45 patients (one per patient) with cSCC skin cancer, from 6 medical centers - Bonn, Cologne, Dortmund, Munich, Oberhausen and Salzburg taken before administration of immune checkpoint inhibitors treatment. 28 of them were classified as responders, showing partial response (PR) or complete response (CR), i.e. tumor shrinkage. 17 patients were classified as non-responders, showing stable disease (SD) or progressive disease (PD) states. More specifically, CR means disappearance of all lesions; PR: 50% or more in decrease of total tumor size; SD: <50% decrease and/or <25% increase of one/several tumor lesions; PD: >25% increase of one/several tumor lesions or new lesions. The classification was determined by a dedicated review of the clinical and radiological imaging by at least two observations not less than 4 weeks apart (following World Health Organization handbook for reporting results of cancer treatment). This collection of classified WSIs represents our fourth dataset, called CPI (see **Data availability statement**).

We used Histo-Miner to extract tissue representative features from the WSIs of CPI dataset. Then, we trained and evaluated an XGBoost classifier [23] for the task of classifying patients in their two categories based on the feature vectors. We evaluated XGBoost classifier through 3 fold cross-validation of 2 splits, train and test (train containing 2/3rd of the data), and performed feature selection on the training split within the cross-validation runs. Due to the limited number of

samples in the dataset (45) we could not perform hyperparameter search within nested cross-validation so we kept the default set of hyperparameters for all the cross-validation runs. Similarly, the low number of folds is constrained by the low number of samples. In fact, having too small validation folds would increase variance in evaluation. We used minimum Redundancy - Maximum Relevance (mRMR) feature selection method [24] which is designed to find the smallest relevant subset of features (maximum relevance) while preventing highly correlated features to be part of this subset (minimum redundancy). All the 107 features kept for analysis are describing tissue structure, no nucleus morphology features (area, circularity) were included (See **Tissue Analyser** section in **Methods** for an in-depth description of the features).

To know how many features to keep we calculated the average balanced accuracy for each training keeping $N \in [1, 107]$ features. We kept $N_{\text{best}} = 19$ features which corresponded to the best mean balanced accuracy across all runs. Even if the number of selected features $N_{best}$ is the same for each run in the case of mRMR, the selection of features can vary. Following Eq 2, we identify the most representative features by selecting those with the highest occurrence counts $c_{f_k}$ across the selected feature sets in each cross-validation run. In the event of ties, we favor features with higher rankings in their respective sets. To do so we first record the position of the feature in its set, its rank, and calculate its pre-score as $10^{N_{best}-rank}$, so a feature ranked first would have the highest pre-score. Then we add all the pre-scores for each cross-validation groups for a given feature and take the log of this sum to obtain the final score $s_{f_k}$.

$$A = \text{concat}\{fv_{split_1}, fv_{split_2}, ..., fv_{split_L}\} \tag{2}$$

$$c_{f_k} = \sum_{l=1}^{L} \sum_{n=1}^{N_b} \delta(A_{l,n}, f_k)$$

$$s_{f_k} = \log(\sum_{l=1}^{L} 10^{N_{best}-rank(f_k)_{split_l}})$$

where $fv_{split_l} = [f_1, f_2, ...f_{N_b}]_{split_l}$ the vector of $N_{best}$ selected features from split $l \in [1, L]$ of the cross-validation, $rank(f_k)_{split_l}$ the rank of feature $f_k$ selected from split $l \in [1, L]$ and $\delta$ the Kronecker delta.

Notably, mRMR prevents redundancy in feature selection, but aggregation of features from all cross-validation runs can include highly correlated features.

## Results

### Nuclei segmentation and classification evaluation

Accurate segmentation and classification of nuclei is necessary, to ensure the quality of downstream analyses involving features based on cell nuclei. We used Panoptic Quality metric [14] to evaluate segmentation, as it has been shown that it is better suited for evaluation of instance segmentation than DICE2 (aggregation of DICE score for each instance) [25] or aggregated Jaccard Index (AJI) metrics [26]. Indeed DICE2 is oversensitive to small changes in the prediction and AJI is over-sensitive to failed detections as shown in [14]. Following its definition, Panoptic Quality also assess detection performance.

We compare our cell nuclei segmentation and classification model to CellViT, current state of art model for segmentation of cells in H&E-stained WSIs [27]. CellViT was originally trained on Pannuke dataset [15] a commonly used and recognized dataset for panoptic segmentation on H&E stained tissues. Pannuke contains diverse types of cancer but only few skin cancer images, and without distinction of the type of skin cancer. Additionally, the dataset does not include granulocytes and plasma cells and the Pannuke-pretrained CellViT is not able to detect them.

We trained both models on NucSeg dataset including 5,968 patches of size 540x540 pixels with 70% overlap. Our validation set contained 848 patches of size 540x540 pixels with 70% overlap (see **Method** section for more details on training dataset and training procedure). No tiles were overlapping between the train and validation sets. We compared segmentation maps from SCC Hovernet and CellViT models applied to the same validation set in Fig 4a. The segmentation and classification results of each cell class are compared one by one.

We also compared the 3 models on the classification task, considering only detected and paired cells (prediction and groundtruth). We evaluate F1 score for each class (Fig 4a). CellViT-SAM-B is the best classifier for 3 classes and SCC Hovernet for the 2 others, average of F1 over all classes is 0.825 for $CellViT_{256}$, 0.846 for CellViT-SAM-B and 0.832 for SCC Hovernet. A confusion matrix of SCC Hovernet classification on the validation set is shown in Fig 4b.

Overall, SCC Hovernet slightly outperforms CellViT-SAM-B in segmentation and detection tasks but slightly underperforms CellViT-SAM-B in solely classification task. Nevertheless SCC Hovernet has 3 times less parameters [28] and, with its convolutional neural network (CNN) architecture, is lighter than CellViT-SAM-B, which make it easier to use for training and inference.

To additionally validate Histo-Miner, we performed immunohistochemistry (IHC) for Myeloperoxidase (Mypo; granulocytes), CD3 (lymphocytes), CD79a (plasma cells), CD10 (stromal cells) and p40 (tumor cells), as well as H&E staining on adjacent slides from the same tissue blocks. In 6 different cSCC tumors we selected 7 to 11 representative ROIs (750μm x 750μm) on H&E-stained image and predicted the number of cells of each type using Histo-Miner (Fig 4c). The same regions in IHC images were classified by a board certified dermatopathologist into 4 groups (- , +, ++, +++) based on the level of IHC positivity. Comparing the number of predicted cells across IHC positivity groups, we observed a significant association of the number of predicted cells per cell type and IHC positivity of the appropriate marker (Fig 4d).

### Tumor segmentation evaluation

To evaluate the performance of SCC Segmenter model, we used intersection over union (IoU) between the predicted tumor regions and the groundtruth of tumor regions. The model was trained on randomly chosen 115 slides of TumSeg dataset (see **Method** section for detailed description of training procedure). The validation set contained 29 slides of the dataset. A test set composed of 32 slides from 25 patients (patients not present in the training and validation sets) was used for evaluation. No data selection was performed to generate the test set. This test set is publicly available (see **Data availability statement**). The model achieves on the test set: segmented foreground intersection over union $IoU_{tissue} =$ 0.852, mean intersection over union mIoU = 0.907, accuracy of foreground segmentation $Acc_{tissue} = 0.892$, and average accuracy mAcc = 0.969.

### Application of Histo-Miner to predict response of cSCC patients to immunotherapy

To demonstrate the usability of Histo-Miner in a clinical scenario, we applied it to predict outcomes of patients treated with immune checkpoint inhibitors using anti-PD1 antibodies. Immunotherapy is the most effective treatment for advanced cSCC patients, but half of all patients do not respond and reliable predictive parameters do not yet exist.

The CPI dataset consists of 45 WSIs from cSCC skin cancer of 6 medical centers before they received immune checkpoint inhibitors treatment. Each WSI is taken from a different patient. 28 of them were classified as responders (CR and PR states) and 17 patients were classified as non-responders (SD and PD states) as displayed in Fig 5a. We performed 3 fold cross-validation with XGBoost classifier [23] on the CPI dataset to predict the class of the WSIs.

### Discussion

WSIs contain vast amounts of detailed information and are a rich resource for cancer diagnosis and research. Automated digital pathology methods offer possibilities to efficiently process WSIs, uncovering quantitative information as well as

**(a)**

| | | Methods | | |
|---|---|---|---|---|
| | | CellViT$_{256}$ | CellViT-SAM-B | SCC Hovernet |
| **Number of parameters** | | 46.8M | 146.1M | 52.2M |
| **Class PQ** | Granulocytes | 0.354 | **0.464** | **0.464** |
| | Lymphocytes | 0.524 | 0.521 | **0.578** |
| | Plasma | 0.460 | 0.502 | **0.565** |
| | Stroma | 0.582 | **0.622** | 0.586 |
| | Tumor | 0.603 | **0.650** | **0.650** |
| | $mPQ$ | 0.505 | 0.552 | **0.569** |
| **Class F1** | Granulocytes | 0.825 | 0.844 | **0.899** |
| | Lymphocytes | 0.846 | **0.853** | 0.803 |
| | Plasma | 0.760 | 0.772 | **0.818** |
| | Stroma | 0.797 | **0.824** | 0.743 |
| | Tumor | 0.922 | **0.935** | 0.899 |
| | $meanF1$ | 0.825 | **0.846** | 0.832 |

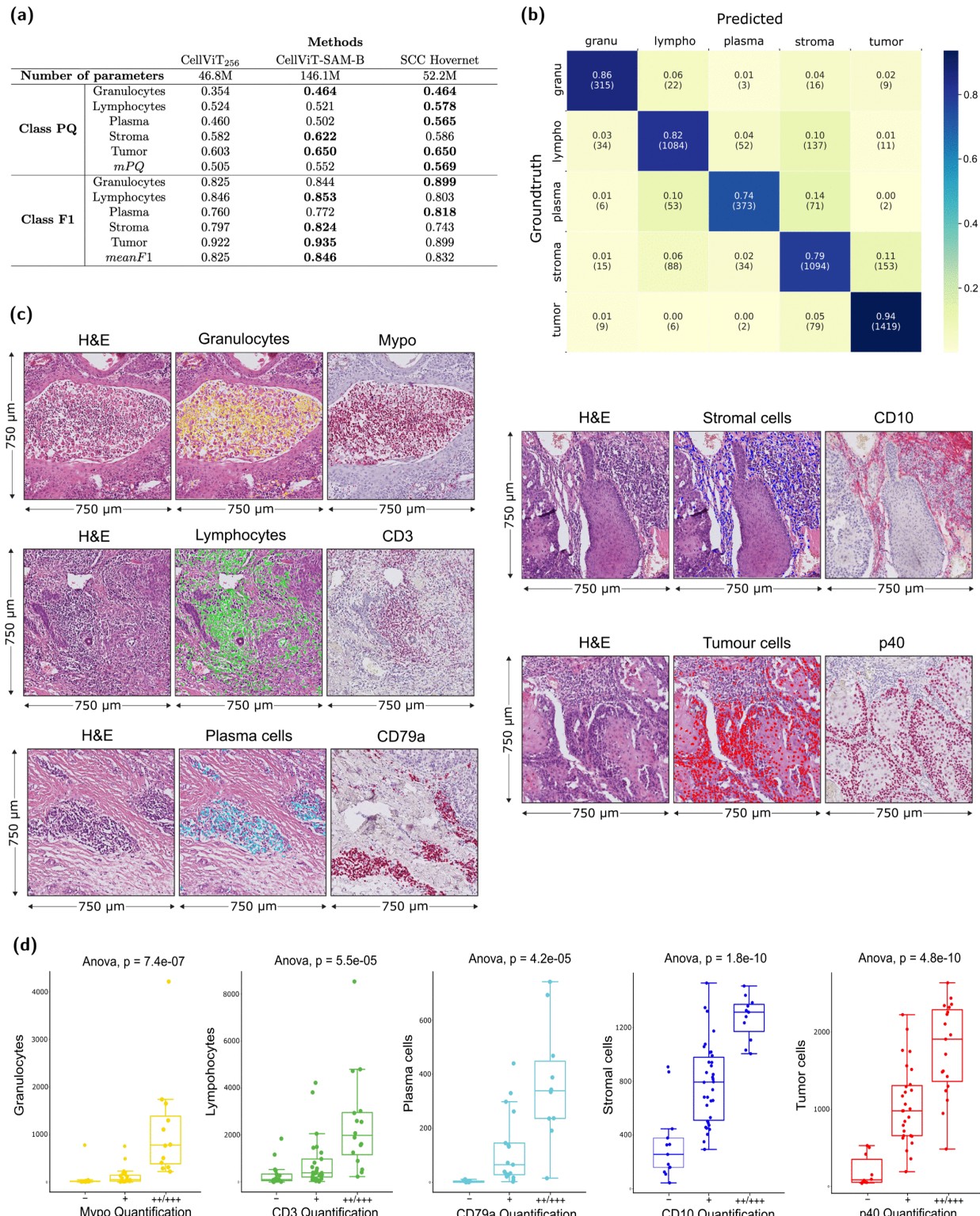

**Fig 4. Validation of SCC Hovernet. (a)** Panoptic Quality for each cell class of SCC Hovernet, light CellViT$_{256}$ and heavy CellViT-SAM-B, all trained on NucSeg. mPQ is the average of PQ for all classes. SCC Hovernet outperforms CellViT$_{256}$ for all classes and outperforms CellViT-SAM-B general mPQ performance. CellViT-SAM-B outperforms SCC Hovernet on general classification performance. Taking into account segmentation, detection, classification tasks and weight of the models, SCC Hovernet is the preferred option. **(b)** Confusion matrix from SCC Hovernet prediction on the validation

set. The most representative class, tumor cells, is accurately classified 94% of the time. The worst classification accuracy is of plasma cells: 74%. **(c)** Examples of validation via immunohistochemistry showing H&E and cell-type predictions (left and middle column) and staining for cell type markers of the same are in a consecutive section (right column) Mypo=Myeloperoxidase. **(d)** Comparison of manual cell type quantification in immunohistochemistry slides (x-axis; Mypo = Myeloperoxidase) and computationally predicted cells (y-axis) in H&E slides.

subtle patterns and features that may be inaccessible or indiscernible to human experts. Moreover, they enable automated cell type classification and detailed description of tissue composition and architecture. These approaches have rarely been applied to non-melanoma skin cancers like cSCC despite the fact that cSCC alone affects more than 1 Million individuals in the USA every year [12]. A major obstacle to development of automated methods for non-melanoma skin cancer slide analysis has been the high similarity between tumor cells and non-tumor skin cells. Here, we present the Histo-Miner pipeline which provides single-cell insights into skin tumor WSIs. Our methods not only generate precise and complete information on the patient biopsy composition, we also demonstrate how this information allows to predict patient outcomes and give interpretable insights into the determinants of these outcomes. Such techniques can therefore lead to discovery of previously unknown diagnostic biomarkers, leading to improved cancer detection, diagnosis, and personalized treatment strategies.

Histo-Miner performs segmentation and classification of cell nuclei using a CNN, SCC Hovernet, trained on our Nuc-Seg dataset, as well as tumor segmentation using a vision transformer, SCC Segmenter, trained on our TumSeg dataset. We compare the performance of segmentation and classification of segmented cells task to state of the art methods, such as CellViT, and show improved Panoptic Quality of our approach. Based on classification and segmentation results, Histo-Miner creates feature vectors describing tissue composition and organization. A Histo-Miner user can choose which features to calculate to best describe their WSIs. Additionally, Histo-Miner models can be trained and adapted to other cancer types.

Given the large size of WSIs, most commonly prediction tasks in digital pathology are performed via multiple instance learning (MIL) approaches [29,30] . Via WSI tesselation and patch embedding, such approaches allow to train the model and perform the prediction on the entire WSIs directly. While it is convenient to train the prediction model end-to-end using patient-level labels that are typically available in the patient records, MIL methods offer only limited interpretability. MIL paired with attention or gradient-based mechanisms [31–35] allow to disentangle which WSI regions are the most predictive, however the content of these highly-predictive regions is typically assessed in a qualitative manner. In contrast, Histo-Miner represents a feature-based approach that allows for a quantitative and systematic identification of tissue characteristics that are important for prediction. Each step of the Histo-Miner pipeline from segmentation to feature selection can be visualized allowing for inspection and interpretation of the prediction results.

We demonstrate the applicability of our pipeline on a cohort of 45 patients treated with immune checkpoint inhibition, which is the major treatment modality for patients with advanced cSCC [13]. Even though our patient cohort was relatively small and collected across 6 medical centers, our pipeline was able to accurately predict CPI response in cross-validation experiments, highlighting its potential for clinical use cases. In addition, our feature-based approach points to the features driving the classification and thus to potential insights into the tumor biology. We identified four features the most predictive of CPI response, which included a higher percentage of lymphocytes within and in the vicinity of tumor regions in responders than non-responders. This result is in line with previous studies showing lymphocyte infiltration as a predictive marker of CPI response in cSCC [36]. Interestingly, we also observed a higher granulocyte to lymphocyte ratio and a higher distance between granulocytes and plasma cells in non-responders than responders. High neutrophil-to-lymphocyte ratio in the blood of cSCC patients has been associated with worse prognosis in general and suggested to correlated with decreased CPI response [37,38], but their ratio in cSCC tissues has not been described before. Similarly, very little is known about plasma cells in cSCC tumors, especially in conjunction with granulocytes. In breast cancer however, patients not responding to CPI showed high degree of granulocytes as measured by CD15 [39] positivity. In mouse

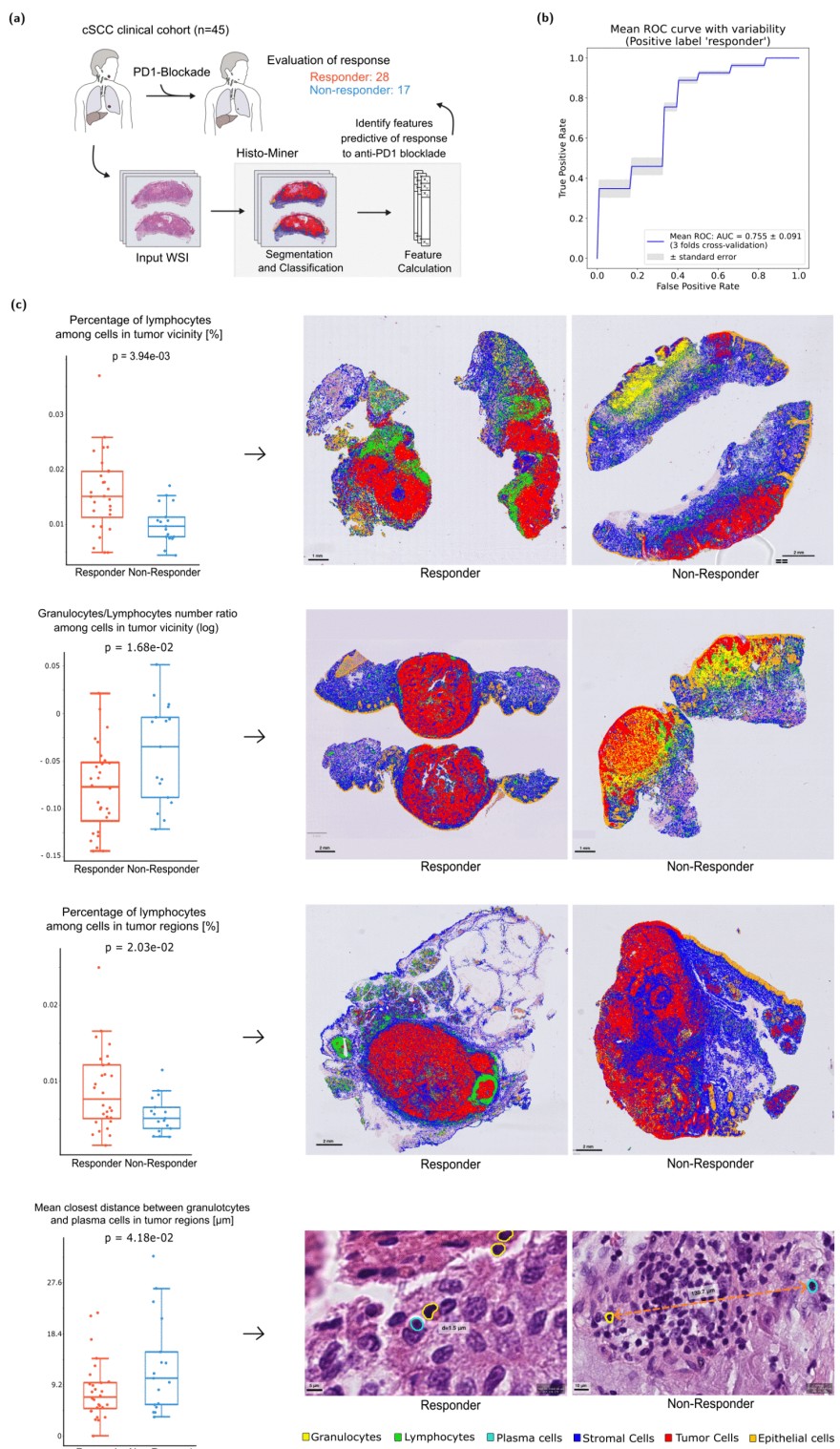

**Fig 5**. **Histo-Miner tested in a clinical scenario: prediction of CPI treatment response. (a)** To test the clinical utility of Histo-Miner we assembled and processed WSIs from in total n=45 cSCC patients before checkpoint inhibition (CPI) therapy. **(b)** Mean ROC curve for the classifier keeping 19 best features and its standard error. The classifier cross-validation folds ROC curves as well as the standard deviation of the mean ROC curve are shown in Fig 6 in S1 Material. **(c)** On the left, box plots of the 4 best features, p-value was calculated using Mann-Whitney U test. On the right, visualization of representative cases for each of the best features. For distance visualization we hide cell classes other than plasma cells and granulocytes.

models of pancreatic and squamous cell lung cancer, depletion of neutrophilic granulocytes led to reduced tumor growth [40,41]. While the interplay between granulocytes and T and plasma cells in cSCC requires experimental validation in future studies, these findings indicate that both cells of the innate (granulocytes) and the adaptive arm (T and plasma cells) of the immune system may play opposing roles in modulating CPI response in cSCC. They thereby highlight one of the strengths of our pipeline, which - in contrast to more coarse approaches - classifies immune cells into 3 subcategories and provides quantitative as well as spatial information about tumor microenvironment to identify relevant factors. It can thereby help to generate novel hypotheses for follow-up investigations.

A limitation of our approach is the requirement of expert-annotated samples (tumor regions, nucleus boundaries, and cell classes) that serve as ground truth for training and testing. It may contain human errors and introduce subjective biases that the models ultimately learn to replicate. In addition, tumor cell recognition using an additional tumor region segmentation model might lead to individual cancer cells outside of the predicted tumor regions as well as non-neoplastic epithelial cells within the predicted tumor regions being misclassified.

A difficulty we faced is the distinction between non-neoplastic epithelial and tumor cell nuclei. For the published pre-trained models we tested, morphologies of those two cell types were indistinguishable in a skin H&E-stained sample if considered in isolation. Context, such as e.g. cell localization and neighborhood, allow to distinguish them from one another. Here the combination of two deep learning models - one for cell segmentation and classification and one for tumor region segmentation - allowed us to discriminate between the two cell classes. Further studies should focus on designing a cell classifier that is able to distinguish all 6 cell classes (granulocytes, lymphocytes, plasma cells, stromal cells, tumor cells, non-neoplastic epithelial) without using an additional tumor region classifier. Such model should incorporate a broader context including surrounding cells in a patch in the prediction process.

## Supporting information

**S1 Material. This supporting document contains all supplementary tables and figures cited in the main text.** It includes the following sections:

- Poor precision in tumor and healthy epithelial detection from state-of-the-art pretrained models.
- SCC Hovernet loss functions.
- Hyperparameters grid search for SCC Segmenter.
- Probability of distance overestimation.
- List of all features from Tissue Analyser.
- Ranking of features after cross-validation with mRMR selection.
- ROC curves of the best-feature classifier across CV folds.
- Stain variability for the different cohorts of TumSeg dataset.

(PDF)

## Acknowledgments

The authors would like to thank Christian Knetschowsky for the annotations of segmentation maps of NucSeg dataset, and to thank Alfred Kirsch for the detailed corrections and extensive verifications brought to the calculation of the probability of closest distance overestimation.

## Author contributions

**Conceptualization:** Lucas Sancéré, Carina Lorenz, Johannes Brägelmann, Katarzyna Bozek.

**Data curation:** Lucas Sancéré, Carina Lorenz, Doris Helbig, Johannes Brägelmann.

**Formal analysis:** Lucas Sancéré.

**Funding acquisition:** Johannes Brägelmann, Katarzyna Bozek.

**Investigation:** Lucas Sancéré, Carina Lorenz.

**Methodology:** Lucas Sancéré.

**Project administration:** Lucas Sancéré, Doris Helbig, Johannes Brägelmann, Katarzyna Bozek.

**Resources:** Doris Helbig, Oana-Diana Persa, Sonja Dengler, Alexander Kreuter, Martim Laimer, Roland Lang, Anne Fröhlich, Jennifer Landsberg.

**Software:** Lucas Sancéré.

**Supervision:** Johannes Brägelmann, Katarzyna Bozek.

**Validation:** Lucas Sancéré.

**Visualization:** Lucas Sancéré, Carina Lorenz.

**Writing – original draft:** Lucas Sancéré, Johannes Brägelmann, Katarzyna Bozek.

**Writing – review & editing:** Lucas Sancéré, Carina Lorenz, Johannes Brägelmann, Katarzyna Bozek.

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
