## [Decision Letter · Decision Letter 0]

28 Oct 2025

PCOMPBIOL-D-25-01576

Histo-Miner: Deep Learning based Tissue Features Extraction Pipeline from H&E Whole Slide Images of Cutaneous Squamous Cell Carcinoma

PLOS Computational Biology

Dear Dr. Sancéré,

Thank you for submitting your manuscript to PLOS Computational Biology. After careful consideration, we feel that it has merit but does not fully meet PLOS Computational Biology's publication criteria as it currently stands. Therefore, we invite you to submit a revised version of the manuscript that addresses the points raised during the review process.

Please submit your revised manuscript within 60 days Dec 28 2025 11:59PM. If you will need more time than this to complete your revisions, please reply to this message or contact the journal office at ploscompbiol@plos.org. Please include the following items when submitting your revised manuscript:

We look forward to receiving your revised manuscript.

Kind regards,

Stacey D. Finley, Ph.D.

Section Editor

PLOS Computational Biology

**Additional Editor Comments :**

The reviewers acknowledge the utility of the approach and that the manuscript is well written. However, some details and explanation are needed to improve clarity and allow for reproducibility. These points should be addressed in a revised manuscript.

**Journal Requirements:**

At this stage, the following Authors/Authors require contributions: Lucas Sancéré, Carina Lorenz, Doris Helbig, Oana-Diana Persa, Sonja Dengler, Alexander Kreuter, Martim Laimer, Anne Fröhlich, Jennifer Landsberg, Johannes Brägelmann, and Katarzyna Bozek. Please ensure that the full contributions of each author are acknowledged in the "Add/Edit/Remove Authors" section of our submission form.

5) We have noticed that you have uploaded Supporting Information files, but you have not included a list of legends. Please add a full list of legends for your Supporting Information files after the references list.

Potential Copyright Issues:

i) Figures 1d, and 5a. Please confirm whether you drew the images / clip-art within the figure panels by hand. If you did not draw the images, please provide (a) a link to the source of the images or icons and their license / terms of use; or (b) written permission from the copyright holder to publish the images or icons under our CC BY 4.0 license. Alternatively, you may replace the images with open source alternatives. See these open source resources you may use to replace images / clip-art:

7) Please amend your detailed Financial Disclosure statement. This is published with the article. It must therefore be completed in full sentences and contain the exact wording you wish to be published.

8) Please ensure that the funders and grant numbers match between the Financial Disclosure field and the Funding Information tab in your submission form. Note that the funders must be provided in the same order in both places as well. Currently, this information "ODP received funding from the Bavarian Cancer Research Center(BZKF) and Deutsche Stiftung fur Dermatologie" is missing from the Funding Information tab.

9) Please amend your 'Competing Interests' statement, and declare all competing interests beginning with the statement "I have read the journal's policy and the authors of this manuscript have the following competing interests:"

Note: If there are no competing interests to declare, please state "The authors have declared that no competing interests exist".

**Reviewers' comments:**

Reviewer's Responses to Questions

Reviewer #1: Histo-Miner is a novel deep learning-based pipeline designed for the analysis of Whole-Slide Images (WSIs) of skin, specifically targeting the challenge of non-melanoma tumor cell classification, where data scarcity and high morphological similarities between tumor and healthy epithelial cells pose difficulties. The pipeline integrates convolutional neural networks and vision transformers for nucleus segmentation and classification, as well as tumor region segmentation. A key refinement step is implemented where nuclei predicted as tumor cells but located outside predicted tumor regions are reclassified as healthy epithelial, leveraging broader tissue context to overcome these morphological ambiguities. Histo-Miner introduces two new cSCC datasets and demonstrates state-of-the-art performance across multiple tasks. Importantly, it shows potential in predicting cSCC patient response to immunotherapy, highlighting key predictive cellular features such as lymphocyte percentages and granulocyte-lymphocyte ratios, and is designed for generalizability to other cancer types and datasets.

Overall, the study presents a promising and well-structured pipeline with clear potential for advancing the analysis of WSIs in cSCC and beyond. However, several details regarding the methodology and evaluation remain unclear and should be clarified to ensure the reproducibility of the study and to allow a proper assessment of the robustness and generalizability of the approach.

2.1 Histo-Miner pipeline description. The pipeline description mentions two types of preprocessing involving downsampling, tiling, and color normalization. However, important details are missing to ensure reproducibility. Could the authors clarify whether tiles are extracted only from tissue regions or from the entire WSI? If tiles are extracted from the whole slide, are patches without tissue filtered out afterwards? The method for color normalization is not fully described, and it would be helpful to clarify how it addresses variability in staining across samples.

2.2 NucSeg and TumSeg datasets descriptions. In the dataset generation section, it is stated that class labels, segmentation masks, and tumor regions were manually annotated by two pathology experts. However, it remains unclear how the annotations from the two experts were combined, and what measures were taken to ensure consistency and reliability of the annotations across the dataset.

2.4 Training SCC Hovernet. Regarding the reported split of 5,968 patches for training and 848 patches for validation from the NucSeg dataset, could the authors clarify whether the patch division was performed at the patient level, ensuring that patches from the same patient were kept within the same split? This is important to avoid potential data leakage and overestimation of model performance.

Could the authors clarify how the validation set was used for model selection? In particular, was early stopping applied, and if so, how was it configured? Additionally, was cross-validation considered to better assess the robustness and generalizability of the results?

This section reports that performance of the resulting models was highest when first pre-trained on ImageNet-21k and subsequently on the non-curated H&E nucleus dataset before fine-tuning. However, these details seem more appropriate for the sections dedicated to Results and/or Discussion rather than the methodological description

2.5 Training SCC Segmenter. The description states that data augmentation was applied, consisting of random resizing, random left–right flipping, and normalization of the images based on the mean and standard deviation of ImageNet-1k pixel values. However, it seems that the explanation mixes data augmentation with normalization, and this should be clarified. In addition, it would be important to specify how the data augmentation was applied (e.g., whether augmentations were always applied or with a certain probability, and whether all transformations were used simultaneously or independently.

The model was trained on 115 randomly chosen slides from the TumSeg dataset, with the remaining 29 slides used for validation. Could the authors clarify whether this split was performed at the patient level, ensuring that slides from the same patient were not included in both sets?

The accuracy estimation was performed on the validation set due to the limited number of slides and the absence of an independent test set. Could the authors clarify whether the validation set was also used to select the best model? Given the small dataset size, cross-validation is recommended to provide a more reliable estimation of model performance and to ensure that the results are not specific to a particular data split, but consistent across different splits.

2.6 Tissue Analyser. Regarding the cell classification refinement, it is mentioned that nuclei predicted as tumor cells outside predicted tumor regions are reclassified as healthy epithelial. This step is crucial to address the morphological similarity between tumor and healthy cells. Could the authors clarify how the robustness of this refinement is ensured against potential inaccuracies in tumor region predictions by the SCC Segmenter? For instance, if a region is incorrectly classified as non-tumor, how does this affect the reclassification of genuinely tumor nuclei within that region? Are confidence thresholds or probability scores used to mitigate such errors?

2.7 Feature selection The subsection on feature selection in the case study appears under Methods but mixes methodological description with reporting of results. It would be helpful if the authors clearly separate the description of the feature selection procedure from the results obtained, to improve clarity and reproducibility.

Reviewer #2: This manuscript presents Histo-Miner, a deep learning-enabled pipeline for analyzing whole-slide images (WSIs) of cutaneous Squamous Cell Carcinoma (cSCC). The pipeline incorporates a trained convolutional neural network for the segmentation and classification of nuclei and a trained vision transformer for the segmentation of tumor areas. After validating the accuracy of Histo-Miner with manually annotated class labels and segmentation masks, the framework creates a compact feature vector summarizing tissue morphology and cellular interactions. To demonstrate its clinical applicability, the authors utilized this pipeline to predict immunotherapy response in 45 cSCC patients based on their pre-treatment WSIs.

Overall, the manuscript is well-written. However, I would like to suggest the following major revisions before getting it published:

- On Page 2, line 70, it would be beneficial if the authors could provide a few references for this claim.

- Zenodo repository links in supplementary materials are not functional. I believe they should be formatted as https://zenodo.org/records/13986860

- On Page 6, lines 163, 179, and 180, the term "resolution" was used instead of "magnification" when referring to microscope magnification settings. It would be helpful if the authors either clarify the actual resolutions (not magnifications) or consistently use the term "magnification" throughout these lines.

- I could not find evidence that the developed method was compared against blind test samples (WSIs). If this comparison was not performed, would it be possible to analyze the accuracy of the technique using blind test samples?

- On Page 4, line 123, the authors mention training and testing the models using manually annotated datasets. It would be valuable if the authors could elaborate on the potential problems of manual labeling and their impact on model accuracy in the discussion section.

Reviewer #3: Summary

The paper introduces a deep learning-based pipeline for the analysis of skin Whole-Slide Images (WSIs), called Histo-Miner. To this aim, the Authors generated new datasets of cutaneous Squamous Cell Carcinoma (cSCC). Histo-Miner is presented as a good predictor for nucleus segmentation, nucleus classification, and tumor region segmentation, achieving results comparable to the state of the art, with improvements in the classification of healthy tissue. Moreover, the work also introduces the use of the tool with a predictive objective for immunotherapy response.

Overall Quality Statement

The work appears to be of very high quality. It presents an interesting tool with strong potential both in image recognition and in its integration with mathematical models, particularly regarding response prediction. The material is clearly presented, with a good balance between text and supplementary material. The flow of the exposition is clear and well documented, including information on the pipeline, practical examples, validation of results, and presentation of possible further applications. Although my expertise is more on the side of application than on the technical aspects of the employed methodologies, I consider this manuscript to be of high value, both for its technical quality and for the usefulness of its content.

Minor Observations

R133:

“After determining tumor areas using SCC Segmenter, the results of the SCC Hovernet cell nuclei classification are updated to add a new cell class as follows: all the nuclei predicted as tumor cells outside of the predicted tumor regions are reclassified as healthy epithelial.”

Q. Could you please discuss the potential risk of omitting information related to invasive tissues or metastatic initiation? Have you checked whether there is any correspondence, when analyzing unresponsive tissues, with tumors showing a large number of cells outside predicted tumor regions?

R144:

“For every pair of cell classes X and Y, we also calculate the average distance of the closest cell of class Y to a cell of class X inside the tumor regions.”

Q. Since later in the manuscript the Authors state that they evaluate local cell densities, I would find it more natural to suggest considering the average density instead (i.e., the average number of cells of class Y around cells of class X within a given sensing radius). This approach would also help address boundary effects, as density can be evaluated relative to the proportion of the sphere lying within the tissue, thereby avoiding singularities. Could you please provide an explanation for this choice?

R176:

“To build a tumor segmenter algorithm, we additionally assembled 144 WSIs of 125 cSCC patients from 3 medical centers – Bonn, Cologne, and Munich.”

Q. It would be useful to include information about data uniformity, since the dataset originates from different centers.

R220:

“We list loss functions in Supplementary Eq. 1–4.”

Q. While it is functional to place technical information in the supplementary material, a short descriptive summary in the main text would be valuable. For instance, a brief explanation of which elements the loss functions account for, and whether these are considered as absolute values or in terms of slope.

R253:

“The accuracy estimation was performed on the validation set due to the limited number of slides and lack of an independent test set.”

Since accuracy was evaluated on the validation set rather than an independent test set, the reported performance may be overestimated. The limited number of slides further reduces the reliability of this estimation. I would suggest either cross-validation or the use of an external validation cohort to better assess generalizability. Could you clarify whether the lack of independence arises from the nature of the dataset itself, or from reliance on data augmentation?

**Have the authors made all data and (if applicable) computational code underlying the findings in their manuscript fully available?**

Reviewer #1: Yes

Reviewer #2: Yes

Reviewer #3: Yes

PLOS authors have the option to publish the peer review history of their article (what does this mean?). If published, this will include your full peer review and any attached files.

Reviewer #1: No

Reviewer #2: No

Reviewer #3: No

**Figure resubmission:**
---

## [Decision Letter · Decision Letter 1]

9 Jan 2026

Dear Mr. Sancéré,

We are pleased to inform you that your manuscript 'Histo-Miner: deep learning based tissue features extraction pipeline from H&E whole slide images of cutaneous squamous cell carcinoma' has been provisionally accepted for publication in PLOS Computational Biology.

Best regards,

Stacey D. Finley, Ph.D.

Section Editor

PLOS Computational Biology

Reviewer's Responses to Questions

**Comments to the Authors:**

Reviewer #1: The authors have satisfactorily clarified the issues raised in the review and followed the suggested recommendations.

Reviewer #2: The manuscript was already well-written and now it is ready-to-publish after these major revisions.

Reviewer #3: I would like to thank the Authors for the work carried out during the revision of the manuscript and for the careful attention given to the Reviewers’ comments.

I have reviewed your responses to the Reviewers’ comments and the revisions made to the manuscript. I consider the corrections implemented in response to the other reviewers’ suggestions, as well as the changes introduced in line with my own remarks, to be appropriate and well justified.

In light of this, I am satisfied with both the revisions made to the manuscript and the responses provided.

**Have the authors made all data and (if applicable) computational code underlying the findings in their manuscript fully available?**

Reviewer #1: Yes

Reviewer #2: Yes

Reviewer #3: Yes

PLOS authors have the option to publish the peer review history of their article (what does this mean?). If published, this will include your full peer review and any attached files.

Reviewer #1: No

Reviewer #2: **Yes:** Cagatay Isil

Reviewer #3: No

---

## [Editor Report · Acceptance letter]

PCOMPBIOL-D-25-01576R1

Histo-Miner: deep learning based tissue features extraction pipeline from H&E whole slide images of cutaneous squamous cell carcinoma

Dear Dr Sancéré,

I am pleased to inform you that your manuscript has been formally accepted for publication in PLOS Computational Biology. Your manuscript is now with our production department and you will be notified of the publication date in due course.

With kind regards,

Anita Estes
